# Exploring the Expression of the «Dark Matter» of the Genome in Mesothelioma for Potentially Predictive Biomarkers for Prognosis and Immunotherapy

**DOI:** 10.3390/cancers15112969

**Published:** 2023-05-29

**Authors:** Emanuela Felley-Bosco

**Affiliations:** Laboratory of Molecular Oncology, Department of Thoracic Surgery, Zürich University Hospital, 8091 Zurich, Switzerland; emanuela.febo@gmail.com

**Keywords:** mesothelioma, transposable elements, transcriptome

## Abstract

**Simple Summary:**

Endogenous retroviruses (ERVs) are integrated retroviral elements that cover 8% of the human genome, representing thereby four-fold the fraction of the genome encoding for protein-coding genes. Overall, their expression represents up to 0.5% of all non-rRNA transcripts in mesothelioma, and some are specifically re-activated. This part of the genome has been considered until recently “junk DNA” or “DNA dark matter”. However, analysis of ERV expression has recently gained attention in several cancers, especially in the context of immunotherapy. In this review, our aim is to raise the curiosity of non-specialists by illustrating the potential of exploring the expression of “DNA dark matter” for prognosis and immunotherapy, potentially as predictive biomarkers in mesothelioma.

**Abstract:**

Recent high-throughput RNA sequencing technologies have confirmed that a large part of the non-coding genome is transcribed. The priority for further investigations is nevertheless generally given in cancer to coding sequences, due to the obvious interest of finding therapeutic targets. In addition, several RNA-sequencing pipelines eliminate repetitive sequences, which are difficult to analyze. In this review, we shall focus on endogenous retroviruses. These sequences are remnants of ancestral germline infections by exogenous retroviruses. These sequences represent 8% of human genome, meaning four-fold the fraction of the genome encoding for proteins. These sequences are generally mostly repressed in normal adult tissues, but pathological conditions lead to their de-repression. Specific mesothelioma-associated endogenous retrovirus expression and their association to clinical outcome is discussed.

## 1. Introduction: Why It Is Important to Explore Expressed Genome in Mesothelioma

Malignant mesothelioma (reviewed in [1,2]) is a rare cancer with dismal prognosis. It arises in the mesothelium, a tissue with mesodermal origins covering the lungs (pleural mesothelioma), peritoneal cavities (peritoneal mesothelioma), the sacs surrounding the heart (pericardial mesothelioma) and the testes (tunica vaginalis mesothelioma). Exposure to asbestos has been since long-identified as a cause of mesothelioma in the seminal experiments of Wagner [3]. Although the use of asbestos has been banned in several countries, there are several developing nations that continue to use asbestos [1], and the incidence of mesothelioma is still on the rise.

The genetic alteration of tumor suppressors such as *CDKN2A* and *CDKN2B,* as observed in many other cancers, and alterations in *BRCA-associated protein 1* (*BAP1*) are frequently observed in mesothelioma tumors (reviewed in [1] and [2]). Another frequently mutated pathway is the NF2/Hippo signaling pathway. Additional less frequent alterations are observed in *TP53*, in *TERT* promoter and in genes involved in RNA metabolism. Of note, recent studies have suggested that alterations in epigenetic and splicing regulators [4,5] may represent a novel hallmark of cancer.

Interestingly, for the past 11 years, we have known that 75% of the human genome is transcribed into RNAs, but contrary to what is observed in bacteria, only 2% of these transcripts are translated into proteins [6]. The discovery that a large part of what had been previously called “junk DNA” is actively transcribed and carries out crucial functions inspired the concept of “RNA as epicenter of genetic information” [7], where RNA are the most-influencing molecules in cellular function in eukaryotes. This is supported by the observation that the transcriptome seems to be the best prognostic factor in several cancers [8].

In this review, we focus on the expression of transposable elements (TE), in particular endogenous retroviruses in mesothelioma. Indeed, although they are frequently not investigated in transcriptomics studies because of their repetitive nature, we observed that their expression increases upon mesothelioma development [9] and is associated with better overall survival [10].

## 2. Endogenous Retroviruses Are the TE with Highest Variation in Mesothelioma

TEs are DNA sequences that move from one location on the genome to another, and they are also known as “jumping genes”. TEs were discovered more than 70 years ago in corn by Barbara McClintock [11]. Then, it was suggested that repeated DNA sequences may affect gene expression through regulatory factors binding sites in their sequences [12]. The total length of these sequences exceeds by a factor greater than 20 that of protein-coding exons [13]. Based on the way they re-transpose in the genome, TEs can be grouped into two major classes: Class I retrotransposons and Class II DNA transposons.

Class I retrotransposons relocate via RNA intermediates, which are reverse-transcribed by a reverse-transcriptase enzyme encoded by the retrotransposon. The DNA is then integrated into the genome [14]. Based on the presence or absence of long terminal repeat (LTR) flanking sequences, Class I retrotransposons are separated into LTR-retrotransposons (also known as endogenous retroviruses or ERV) and non-LTR retrotransposons [15]. The latter cover around 32% of the human genome and include long interspersed nuclear elements (LINEs) and short interspersed nuclear elements (SINEs), per [13] (Figure 1a,b).

LINEs are also autonomous elements, while SINEs are non-autonomous retrotransposons that use the retro-transcription machinery of other TEs (LINEs) [16].

Class II DNA transposons can directly relocate autonomously via a cut and paste mechanism. They are active in many lower organisms, including bacteria [17], but they are inactivated and can no longer transpose in humans, although they cover around 3% of the human genome (Figure 1a). However, DNA transposon-derived sequences affect human biology after having been repurposed over evolutionary time. This is the case in recombination activating genes (RAG1 and RAG2), which derive from *Transib* DNA TE. Their transposition mechanism mediates V(D)J recombination in B and T cell precursors [18,19].

ERVs are repetitive elements derived from infection by ancient exogenous retroviruses integrated within the genome of germline cells. They have then transmitted through their offspring [20]. ERVs, however, unlike original retroviruses, are deficient in most transposon functions, and represent defective genomic remnants of the retroviral replication cycle. ERVs comprise a significant part of our genomes, covering 8% [13,21] of the human genome (Figure 1a).

The discovery of human ERVs (HERVs) followed the screening of human genomic libraries using either probes from animal retroviruses or by using oligonucleotides with similarity to virus sequences under low-stringency conditions [22]. HERV length varies from 1.5 to 11 kbp [13]. Full-length HERV proviruses (Figure 1b) include *gag*, which encodes for capside, *pol*, which encodes the machinery necessary for autonomous replication and *env,* which encodes for the envelope glycoprotein, flanked by two LTRs. About 90% of ERVs are present in the human genome as single LTRs, termed “solo LTRs”, which originated from recombination events, lacking all open reading frames (ORFs) [23].

A lot of work has been performed for classifying and naming ERVs [24], particularly down to the level of individual proviral loci [25]; however, ERV classification still remains a considerable problem [26]. ERV sequences can be placed into three classes according to their internal homology to exogenous retroviruses, based on phylogenetic analyses of the conserved regions of the *pol* gene. ERVs clustering with gamma- and epsilonretroviruses are termed Class I; those clustering with lentiviruses, alpha-, beta-, and deltaretroviruses are termed Class II; and those that cluster with spumaviruses are termed Class III, which is the most abundant and also may be the most ancient endogenous retrovirus [24] (Table 1).

MaLR (mammalian apparent LTR retrotransposon) elements, which lack a detectable *pol*-related sequence, are also considered “Class III” due to the slight homology of some members to Class III *gag* sequences.

The relative proportions and abundances of Class I, II and III ERVs, as well as MaLR, differ among species, although they are present in all mammalian species tested [27]. For example, Class II ERVs are significantly expanded in mice.

We observed that the expression of ERVs increases during mesothelioma development in mice exposed to asbestos [9]; therefore, our next step was to investigate human samples by comparing available tumor RNA-sequencing (RNA-seq) data from two mesothelioma cohorts: TCGA [28] and Bueno’s [29], or the mesothelioma primary culture cell lines of the FunGeST series [30,31], with human-embryonic-stem-cell-derived mesothelium [32] used as the non-tumor control. Indeed, for the time being, the only RNA-seq data available for normal mesothelial cells were provided in a single-cell mRNA-seq study in the context of a human cell atlas [33]. The polyA RNA-seq protocol that was used has been shown to fail to detect several classes of repeat RNA [34]. We observed that ERVs are the most abundant TEs upregulated or downregulated in mesothelioma patient datasets when compared to mesothelium. ERVs represent on average 74% of the more-than-two-fold upregulated and 48% of the more-than-two-fold downregulated TEs.

In human mesothelioma, we observed that the ERV1 family, which belongs to ERV Class I, has the highest number of upregulated or downregulated loci in ERVs when compared to mesothelial cells [10].

Few ERV-encoded genes, including Class I, are well annotated in databases and can be explored in classical analysis. One of these, *env* encoding *ERVMER34-1* (also called *HEMO*), is preferentially translated in mesothelioma cells compared to mesothelial cells [35], indicating that proteins from ERVs are produced. Of note, in a pan-cancer analysis, the expression of *ERVMER34-1* was associated with immune system repression [36].

Within the ERV1 family, there are three ERV sequences, *LTR48B*, *LTR7Y* and *LTR6*, that are upregulated in all mesothelioma datasets compared to mesothelium. Of note, *LTR7Y* is an ERV specifically upregulated in early embryo development [37]; therefore, its upregulation in mesothelioma points to developmental signaling re-activation.

A lower number of changes was also observed in the Class III oldest HERV family HERV-L, which has been estimated to be more than 60 to 70 million years old, and the youngest HERV-K (HML-2) family, which is approximately five million years old [38] and belongs to Class II.

The HERV-K family is closely related to the exogenous mouse mammary tumor virus (MMTV) causing breast cancer in mice and has the greatest coding capability [39]. Many HERV-K ERV are still transcriptionally active [40]. There are more than 1000 HERV-K loci in the human genome; however, most of them are “solo-LTRs” [41]. HERV-K family members are less enriched in mesothelioma when compared to mesothelial precursors [10]. *ERVmap-k48,* also called *HERV-K15* [42] had the same expression levels in mesothelioma and mesothelial cells [10]. *Gag* encoding the *ERVmap-k48* locus is near the housekeeping gene *SSBP1,* and its stable levels in normal vs. tumor tissue might be linked to their coregulation [42].

## 3. ERV and Epigenetic Regulation in Mesothelioma

Various mechanisms, including accumulation of mutations, RNA silencing and epigenetic regulation control ERV expression and activity [43]. HERV expression is repressed in the pre-implantation embryo and is maintained in most adult tissues [44]. Most of the mechanisms of HERV repression have been inferred from observations in mouse ERVs. In mice, ERV transcription is suppressed by DNA methylation and/or repressive histone modifications depending on tissue type [45]. Some ERV are normally expressed in various developmental stages of human embryogenesis, and their activity is regulated by alterations in epigenetic regulation in different human tissues [43,46,47].

DNA methylation, which is used by cells to silence transcription, is one of the primary epigenetic mechanisms by which ERV expression is regulated. Indeed, DNA methylation plays an important role in different stages of embryogenesis and in somatic cells in regulating ERV transcription [43,47,48,49,50]. In humans, some HERV transcripts are detectable in many cell types [51], but most ERVs are heavily methylated in somatic cells [52,53,54,55,56]. We observed that ERVs with higher expression in mesothelioma are the consequence of the demethylation of nearby CpG islands [9,10]. In addition, it is worth noting that some of the ERVs that are specifically overexpressed in mesothelioma are near genes such as, e.g., the *MSLN* locus on chromosome 16, which is especially enriched in ERV expression. Mesothelin expression is driven by both YAP activation [57] and promoter de-methylation [58]. The mouse *Msln* promoter, which has been used to generate transgenic mice expressing large T-antigen upon asbestos exposure [59], contains *Mer54b* TE; therefore, one could make the hypothesis that the asbestos-induced demethylation-driven expression of *Mer54b* contributes to large-T antigen expression in MexTag mice.

Besides DNA methylation, histone modifications also play major roles, particularly in undifferentiated and/or stem cells [26]. SETDB1 (SET domain bifurcated histone lysine methyltransferase 1) is a protein lysine methyltransferase that adds methyl moieties to histone H3 at lysine 9 (H3K9). SETDB1-mediated H3K9 tri-methylation (H3K9me3) is responsible for the silencing of ERVs [60,61,62]. H3K9me3 is deposited on ERVs by the SETDB1/KAP1 complex (KAP1 is also known as TRIM28) [63] (Figure 2a).

Of note in the mesothelioma TCGA patients, the quartile with the lowest KAP1 expression has improved overall survival compared with the patients in the higher quartile [64]. It would be interesting to learn whether this is associated with any change in ERV expression.

KAP1 interacts with Krüppel-associated box zinc finger proteins (KRAB-ZFPs), and it has been proposed that the SETDB1/KAP1 complex is recruited to distinct ERVs by different KRAB-ZFPs [65,66,67] (Figure 2a). Therefore, the level of variation in HERV-associated KZFP can potentially explain the differential expressions of HERV in mesothelioma. For example, the expression of ZNF534, which is associated with pluripotency [68], is higher in sarcomatoid compared to epithelioid mesothelioma [29]. In addition, since increased expression of ERVs is observed upon loss of function of SETDB1 [69], it is likely that the subset of mesothelioma patients with mutated SETDB1 display different levels of ERV expression [28,29,70,71]; however, this remains to be demonstrated. An additional regulator of SETDB1 is the human silencing hub (HUSH) complex, which, although mostly involved in LINE silencing, contributes to the regulation of ERV expression (reviewed in [38]).

Global H3K9me3 levels are also decreased and are accompanied by increased ERV expression downstream from the loss of function of lysine acetyltransferase Tat-interactive protein 60. Indeed, the acetylated lysines resulting from the activity of this enzyme recruit bromodomain-containing protein 4, resulting in the increased expression of H3K9 methyltransferases SUV39H1 and SETDB1 and higher levels of H3K9me3 [72].

Some ERVs have promoter or enhancer activity and are associated with active epigenetic marks [73] such as H3K4me1 (an enhancer mark) and exhibit enhancer activity in reporter assays [74]. The depletion of *Kap1* in mouse embryonic stem cells leads to the loss of the repressive mark H3K9me3 at selected ERVs, but also the gain of enhancer marks H3K27ac and H3K4me1, which correlates with the induction of a subset of nearby genes [75] (Figure 2b).

## 4. HERV and Type 1 Interferon Activation

HERVs have been considered as “junk” DNA without biological functions for a long time. However, some HERVs have been co-opted into physiological roles in the host [76]. The best-described example is the production of syncytin-1, a retroviral protein coded by the *env* gene of a provirus belonging to the HERV-W group, expressed in human trophoblasts [77,78,79,80,81]. In addition, HERVs are important determinants of the pluripotency of human embryonic stem cells and of the reprogramming process of induced pluripotent stem cells [37,82].

We documented that ERV transcripts with increased expression in mesothelioma form double stranded RNA (dsRNA) [9,10]. dsRNA is part of the molecular patterns activating the type-I IFN response, and in the tissue of mice exposed to asbestos, we observed 27 interferon-stimulated genes (ISGs) with an expression higher in tumor samples when compared to the samples from asbestos-exposed mice with no tumors [9]. When we investigated the association of the expression of these ISGs with clinical outcome in mesothelioma patients in the TCGA study [28], we found that overexpression of six of them (*DDX58*, *IFIT2*, *IFIT3*, *IFITM1*, *IRF1*, *RSAD2*), is associated with best overall survival, as it has been previously described for the type I interferon signaling pathway [83]. In human mesothelioma cell lines, the increased expression of ISGs is dependent on the activity of dsRNA sensors and the presence of *interferon B1* gene [10]. The latter is located on chromosome 9, near the *CDKN2A* gene, a gene altered in 50–60% of mesothelioma ([84,85]). The *IFNB1* gene is co-deleted in 70% of mesothelioma with *CDKN2A* deletion [86].

The increased expression of ERV transcripts forming dsRNA structures resulting in the activation of dsRNA sensors is consistent with previous observations of MDA5 activation observed due to increased synthesis of ERV upon suppression of DNA methyltransferases [87,88]. Downstream of MDA5 activation, type 1 interferon signaling is stimulated (reviewed in [89]).

Viral mimicry response is widely associated with the expression of TE in several cancers [90] and in a mesothelioma development model [9]. In agreement with these observations, we observed that human mesothelioma tumors expressing high levels of *ERVmap_1248,* which is the near-full-length ERV with the highest expression in mesothelioma compared to mesothelial cells, show a basal activation of type-I IFN signaling and are associated with longer overall survival (Table 2) in three mesothelioma datasets [10].

The samples with the highest *ERVmap_1248* expression also had the highest inflammatory signature score (based on the expression of *CD8A*, *STAT1*, *LAG3* and *CD274*), which predicts response to checkpoint inhibitors in the CheckMate743 trial, where previously untreated mesothelioma patients received either the immune checkpoint inhibitors anti-PD1 and anti-CTLA-4 or chemotherapy [91]. Future studies should prospectively assess the value of *ERVmap_1248* expression as predictive of sensitivity to immunotherapy including immune checkpoint inhibition. However, for therapies inducing type-I IFN signaling, if the contribution of the production of IFNB1 by tumor cells themselves has a role, it should not be forgotten that some forms of mesothelioma have lost type-I IFN genes, as mentioned above [86].

There are several examples where the expression of ERV has been associated with better outcome and/or response to immunotherapy. The activation of type-I IFN is associated with responses to immune checkpoint inhibitors in clear cell renal cancer, which originates in organs from embryonic-related tissue as mesothelioma [52]. In an urothelial cancer cohort, ERV expression is a better predictor of patient response to anti-PDL1 therapy when compared to type-I IFN signatures [34]. The expression of some well-defined [92] ERVs is associated with both immune activation and immune checkpoint signaling upregulation [93] in clear cell renal cell carcinoma. The expression levels of one of these ERVs, *ERV3-2*, predicted response to single-agent PD-1/PD-L1 antibody, and it has recently been demonstrated that this particular ERV is mostly expressed in immune cells [94], which shows that selected ERVs point to responsive immune cells. In an additional clear cell renal cell carcinoma study, *ERV_4700* (or *ERVmap_4700*) expression predicted response to immunocheckpoint blockade and was associated with translation, meaning the production of neoantigens [95]. In another clear cell renal cell carcinoma study, RNAseq performed on more than 100 formalin-fixed paraffin-embedded patient samples from the two clinical trials (CheckMate-010 and CheckMate-025) identified a link between *ERV_2282* (or *ERVmap_2282*) expression with better overall survival in the group treated with anti-PD-1 [96]. Differences between the different studies in this cancer type have been suggested to be linked not only to RNA sequencing approaches, as previously mentioned, but also to cohort differences and ERV mapping and annotation (reviewed in [97]).

In melanoma patients, high ERV and other TE expression is associated with low-risk tumors with better outcomes, while the opposite was associated with repression of TE [98]. Complete response to anti-PD1 treatment has been shown to be associated with high *ERVmap_2637* expression in melanoma patients [60] and negatively correlated with *KDM5B* expression, which recruits SETDB1. In non-small-cell lung cancer, high expression of the *MER4* ERV is associated with better clinical outcome and efficacy of anti-PD1/anti-PDL1 [99] therapy. In addition, ERV envelope glycoproteins have been recently identified as the dominant anti-tumor antibody target in non-small lung cancer [100].

Therapeutic approaches depending on the type-I IFN signaling pathway have already been implemented in the clinic [101] to treat mesothelioma patients. Preclinical studies have suggested the role of this signaling either to improve sensitivity to chemotherapy or in the sensitivity to oncolytic therapy [102,103].

Because of the correlation between the expression of the selected ERVs and the signature predictive of the response to immunocheckpoint inhibition (Table 2) and type I interferon, ERV expression could be useful to select those mesothelioma patients with epithelioid histotype that have the potential to respond to immune checkpoint inhibition, since this tumor histotype responds less compared to the sarcomatoid histotype [104].

## 5. Conclusions

In this review, we summarized the current knowledge on ERVs and their contribution to mesothelioma phenotypes. Besides being associated with the activation of type 1 interferon, HERVs contribute to the expression of tumor-specific antigens in several cancers including mesothelioma [105]. Indeed, HERVs can function as alternative promoters for nearby genes in tumor cells, and cryptic transcription start sites within HERV can generate aberrant protein-coding mRNAs. HLA-pull-down experiments have confirmed that these proteins act as neoantigens, and it will be of interest if some of these overlap with tumor-cell-derived vaccines currently used in the clinic [106].

In addition, these novel ORFs may lead to uncontrolled cell proliferation [107]. For example, aberrant high activity of LTR THE1B contributes to a high degree of transcription of colony stimulation factor receptor 1 in Hodgkin’s lymphoma [108]. HERV-induced chromosomal translocation is another oncogenic mechanism as has been shown in prostate cancer, where HERV-K_22q11.3 is fused to the oncogene E26 transformation-specific family gene ETV1 [109]. We already mentioned that the choice of RNA-sequencing strategy (polyA enrichment vs. rRNA depletion) introduces a bias in the analysis of TE expression. Novel long-read technologies [110], which allow the capture of full-length sequences, will improve the detection and the annotation of expressed TE, including co-transcripts and chimeric transcripts.

It has recently been acknowledged that classical immunotherapy sensitivity biomarkers such as tumor mutation burden and PD-L1 expression have limited predictive value in mesothelioma immunotherapy (reviewed in [111]). Ongoing studies investigating tumor microenvironment, including single-cell sequencing and novel analysis pipelines for neoantigen discovery, will help address the remaining questions about the importance of the expression of selected ERVs in mesothelioma tumor cells or in the tumor microenvironment and their predictive value for immunotherapy.

Finally, besides ERVs, other TEs such as selected LINE1 are overexpressed in mesothelioma [10], and their contribution to pathology and evaluation as biomarkers has yet to be evaluated.

## Figures and Tables

**Figure 1 cancers-15-02969-f001:**
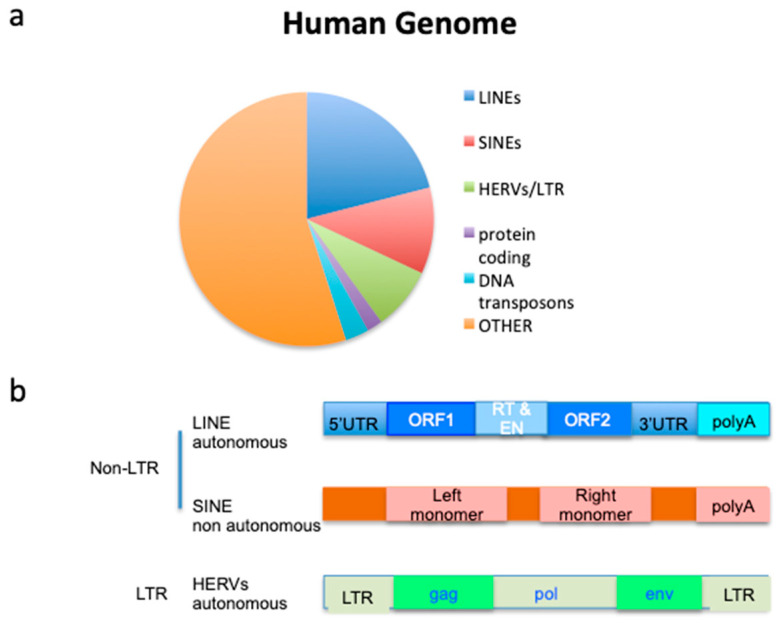
(**a**) Percentage of the human genome covered by transposable elements (TEs) compared to protein-coding sequences. (**b**) The classification of Class I TE. EN: endonuclease; RT: reverse transcriptase.

**Figure 2 cancers-15-02969-f002:**
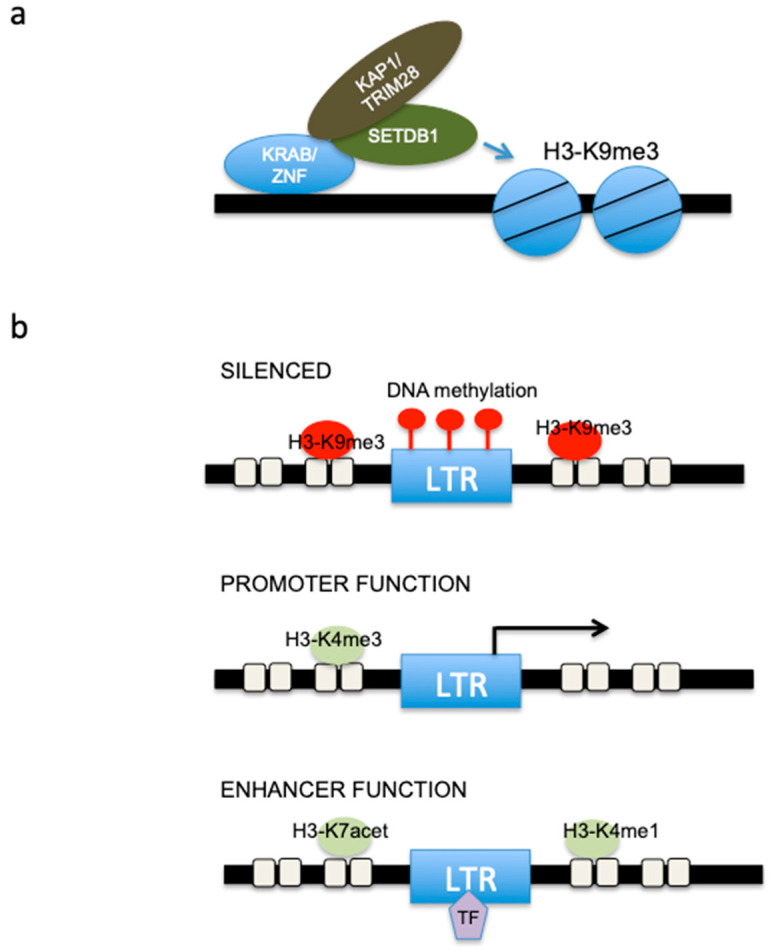
(**a**) A model for SETDB1 recruitment. Shown is a schematic illustrating the current model for KAP1-mediated transcriptional repression. KAP1 is thought to be brought to the genome by interaction with a KRAB-ZNF, which binds specific sites in the DNA. KAP1 in turn has been suggested to recruit the histone methyltransferase SETDB1, which then specifically mediates trimethylation of lysine 9 of histone H3 near the KRAB-ZNF binding sites. (**b**) Upper scheme: most ERVs are epigenetically silenced by DNA methylation and/or repressive histone modifications. Middle scheme: some ERVs function as promoters and are marked by promoter-associated histone modifications. Lower scheme: some ERVs have enhancer activity and are marked by enhancer-associated histone modifications and transcription factors (TF). LTR: long terminal repeat.

**Table 1 cancers-15-02969-t001:** Representative human ERVs.

Class	Related Exogenous Group	Group
I	Gamma	HERV1, HERV-H *, HERV-W *, HERV-E *, ERVMER34-1
II	Beta	HERV-K *
III	Spuma	HERV-L *, MaLR

*: ERVs are also sometimes defined according to the cellular tRNA molecule that is used as a primer for reverse transcription via annealing to an 18 bp binding site sequence. The letter indicates the specific tRNA, e.g., H: His tRNA.

**Table 2 cancers-15-02969-t002:** Human ERVs identified as prognostic or immunotherapy biomarkers in various cancer types.

ERV	Cancer	Biomarker Type	Reference
*ERVmap_1248*	Mesothelioma	Prognostic	[10]
*ERV3-2*	Clear cell renal cell carcinoma	Immunotherapy response	[93,94]
*ERV_4700*	Clear cell renal cell carcinoma	Immunotherapy response	[95]
*ERV_2282*	Clear cell renal cell carcinoma	Prognostic	[96]
*ERVmap_2637*	Melanoma	Immunotherapy response	[60]
*MER4*	Non-small cell lung cancer	Prognostic and immunotherapy response	[99]

## Data Availability

Publicly available datasets were analyzed in this study. These data can be found at zenodo.org: https://doi.org/10.5281/zenodo.7220103.

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
