# Peer review of "Exploring the Expression of the «Dark Matter» of the Genome in Mesothelioma for Potentially Predictive Biomarkers for Prognosis and Immunotherapy"

_cancers, 2023, doi:10.3390/cancers15112969_

Round 1
Reviewer 1 Report
The review explores the expression of the «dark matter» of the genome in mesothelioma for prognostic and immunotherapy potentially predictive biomarkers. The review discusses the mechanisms of epigenetic regulation and the role of endogenous retroviruses (ERVs) in mesothelioma.
The authors provide appropriate and adequate references to related and previous work, giving the reader a clear sense of the current state of the field and how the current work builds on existing knowledge.
The introduction provides a clear and concise explanation of the significance of exploring the expressed genome in malignant mesothelioma. The work is well organized and comprehensively described, making it easy to follow the authors' thought process and reasoning. The article is scientifically sound and not misleading, providing appropriate and adequate references to related and previous work. The English used throughout the review is correct and readable, with no grammatical errors or awkward phrasing. The quality of the English language is high.
Overall, the work is a significant contribution to the field of mesothelioma research, providing valuable insights into the role of expressed genome in this rare and deadly cancer. The introduction sets a clear context for the research, highlighting the importance of understanding the genetic alterations and pathways involved in mesothelioma. The authors' focus on the expression of transposable elements and endogenous retroviruses in mesothelioma is particularly novel and adds to our understanding of the disease.
These articles are relevant to the topic of pleural mesothelioma and miRNA, and I suggest the authors to consider them in their review:
1. Zanellato I, Colangelo T, De Luca G, et al. miRNA profiling in malignant pleural mesothelioma reveals neurotrophin signalling pathway as a potential therapeutic target. Sci Rep. 2017;7:44574. doi: 10.1038/srep44574
2. Pizzimenti S, Toaldo C, Pettazzoni P, et al. miRNA expression analysis in human mesothelial cells exposed to asbestos: evidence of a specific signature. Int J Mol Sci. 2017;18(10):2075. doi: 10.3390/ijms18102075
3. Loreto C, Lombardo C, Caltabiano R, et al. miRNA dysregulation in malignant pleural mesothelioma: miR-16-5p as a new marker of prognosis and survival. Int J Mol Sci. 2020;21(14):5047. doi: 10.3390/ijms21145047
4. Caltabiano R, Puzzo L, Barresi V, et al. Role of miRNAs in malignant pleural mesothelioma. Non-coding RNA Res. 2020;5(4):163-170. doi: 10.1016/j.ncrna.2020.10.001
Reviewer 2 Report
It’s an interesting topic. The author described the endogenous retrovirus elements in the genome, which were not well investigated. The author discussed the potential function for mesothelioma development and immunotherapy efficacy. I have some major comments as follows:
1. In line 132, the author mentioned that ‘there is no RNA-seq data available for normal mesothelium’, which is not really the case. Because there are some GEO datasets, where RNA-seq was done for both normal pleura and mesothelioma tissues, such as GSE2549 (From Bueno’s group), although there are only several normal pleura tissues. Besides, a paper has been published with single-cell RNA sequencing of human pleura ‘https://www.nature.com/articles/s41586-020-2157-4#Sec2, Construction of a human cell landscape at single-cell level’. In addition, scRNA-seq of human pleura has been done in this pre-print paper ‘Single-cell transcriptomic analysis of human pleura reveals stromal heterogeneity and informs in vitro models of mesothelioma’. Therefore, it would be great if the author has a look at ERVs by the datasets. Maybe these data could help to understand the ‘ERVs’ in normal pleural and mesothelioma.
2. Study of ERVs in mesothelioma seems quite rare, while only two studies from the author. So not so many descriptions of ERVs in mesothelioma can be written in a review. Not sure if it’s necessary to write a review about ERVs in mesothelioma.
It will be helpful to add a table to describe the specific ERVs to show whether they can be used for prognostic or immunotherapy biomarkers, based on the author’s work and the study about the functions of ERVs in other types of cancer. The author can predict the potential functions of ERVs in mesothelioma based on research in different types of cancer. In other words, a summary table for sections 3 and 4.
English is good. Some grammar mistakes need to be checked and corrected. For example, in the last sentence 'Finally, besides ERV, other TE such as selected LINE1 are overexpressed in mesothelioma', TE should be TEs (plural).
Reviewer 3 Report
This was a nice study overall about an interesting topic
Overall nice
but as a clinician, I would like in the discussion just how close we are to applying therapeutic targets to mesothelioma, can the authors add any information or new studies preclinical looking at treatment towards the dark matter?
Thank you
Overall nicely written, needs some english editing.
Round 2
Reviewer 2 Report
Thanks a lot for the explanation to address my comments.